# Education and Training Adaptations for Health Workers during the COVID-19 Pandemic: A Scoping Review of Lessons Learned and Innovations

**DOI:** 10.3390/healthcare11212902

**Published:** 2023-11-04

**Authors:** Perla Boutros, Nour Kassem, Jessica Nieder, Catalina Jaramillo, Jakob von Petersdorff, Fiona J. Walsh, Till Bärnighausen, Sandra Barteit

**Affiliations:** 1Heidelberg Institute of Global Health (HIGH), Faculty of Medicine, Heidelberg University Hospital, Heidelberg University, 69120 Heidelberg, Germany; perla.boutros@uni-heidelberg.de (P.B.); nour.kassem@uni-heidelberg.de (N.K.);; 2Africa Health Research Institute (AHRI), Somkhele, Mtubatuba 3935, KwaZulu-Natal, South Africa; 3Department of Global Health and Population, Harvard T.H. Chan School of Public Health, Boston, MA 02115, USA

**Keywords:** SARS-CoV-2, COVID-19, education, medical training, pandemic, adaptation measures

## Abstract

**Introduction:** The COVID-19 pandemic has considerably impacted the clinical education and training of health workers globally, causing severe disruptions to learning environments in healthcare facilities and limiting the acquisition of new clinical skills. Consequently, urgent adaptation measures, including simulation training and e-learning, have been implemented to mitigate the adverse effects of clinical education. This scoping review aims to assess the impact of COVID-19 on medical education and training, examine the implemented adaptation measures, and evaluate their effectiveness in improving health workers’ education and training during the pandemic. **Methods:** Employing the PRISMA-ScR framework and Arksey and O’Malley’s methodological guidance, we conducted a scoping review, systematically searching PubMed, medRxiv, Google, and DuckDuckGo databases to account for the grey literature. The search included studies published between 1 December 2019 and 13 October 2021, yielding 10,323 results. Of these, 88 studies focused on health worker education and training during the pandemic. **Results:** Our review incorporated 31,268 participants, including physicians, medical trainees, nurses, paramedics, students, and health educators. Most studies (71/88, 81%) were conducted in high-income and lower-middle-income countries. The pandemic’s effects on health workers’ clinical skills and abilities have necessitated training period extensions in some cases. We identified several positive outcomes from the implementation of simulation training and e-learning as adaptation strategies, such as enhanced technical and clinical performance, increased confidence and comfort, and an expanded global educational outreach. **Conclusions:** Despite challenges like insufficient practical experience, limited interpersonal interaction opportunities, and internet connectivity issues, simulation training, e-learning, and virtual training have proven effective in improving clinical education and training during the COVID-19 pandemic. Further research is required to bolster preparedness for future pandemics or similar situations.

## 1. Introduction

The COVID-19 pandemic has had a profound effect on global healthcare systems, placing unprecedented demand on health workers. These professionals faced not only increased workloads but also significant shifts in their roles, leading to mental, physical, and economic strain [1]. These challenges underscored the necessity for better support mechanisms and greater recognition of health workers’ efforts [2].

Globally, healthcare systems responded by devising innovative strategies to bolster support for frontline health professionals. This involved infrastructural adjustments and addressing daily challenges presented by the pandemic. For instance, hospitals restructured staff allocations to cater to the surge in COVID-19 patients, introducing new scheduling systems [3] and mobilizing medical students and volunteers [4,5,6]. Digital health approaches, such as telemedicine, became instrumental in facilitating remote patient care [7]. Additionally, digital tools, like electronic triage algorithms based on radiographic findings [8] and staff surveillance systems [9], played a pivotal role in disease containment.

A significant area impacted by the pandemic was the education and training of health workers. Before the pandemic, medical education was characterized by mainly in-person interactions, with theoretical knowledge imparted through classroom lectures and discussions [10] and clinical skills taught through hands-on training [11]. However, the pandemic’s safety protocols necessitated a reimagining of these methods. Education and training programs rapidly implemented new strategies to protect health workers and trainees by reducing the number of people in clinical areas and enclosed spaces. Many in-person activities, including lectures, clinical rotations, and examinations, were suspended [12]. The overwhelming number of COVID-19 patients further strained healthcare resources, diverting attention from trainee education and limiting their hands-on patient care experiences [13]. Limited learning and training opportunities presented significant challenges for health professionals across disciplines, necessitating the development of innovative teaching methodologies to ensure continued learning and skill acquisition [14].

We conducted a scoping review to evaluate the impact of COVID-19 on medical education and training, explore adaptation measures, and assess the effectiveness of these strategies on health workers, leading to novel insights, practices, and innovations regarding health professionals’ education and training during the COVID-19 pandemic.

The primary research questions of the study are as follows: How did the COVID-19 pandemic influence health worker education? What strategies were adopted during this period to enhance both clinical and theoretical training?

The objectives of this scoping review are twofold: (1) to elucidate the ramifications of the COVID-19 pandemic on clinical education and (2) to identify and evaluate the adaptive strategies introduced in response to this.

## 2. Methods

### 2.1. Overview

Given the broad nature of this research question, a scoping review was deemed suitable. This review is grounded in the methodological framework by Arksey and O’Malley [15], which has been further advanced by Levac et al. [16]. The review process encompassed the following five stages: (1) research question identification, (2) the relevant identification of studies, (3) the relevant selection of studies, (4) data charting, and (5) results collection, summarization, and report. The PRISMA-ScR checklist is adopted in Appendix A [17]. We did not conduct a quality appraisal consistent with the nature of a scoping review and in adherence to the guidelines provided by Arksey and O’Malley [15].

### 2.2. Search Strategy

A comprehensive search of the literature was executed on 13 October 2021. We primarily utilized PubMed, a leading biomedical database known for its exhaustive coverage of health-related research. Recognizing the dynamic nature of research, particularly concerning the COVID-19 pandemic, we incorporated medRxiv to access preprints, ensuring the inclusion of the latest studies awaiting peer review. Although Google and DuckDuckGo are not conventional academic databases, they offer access to the gray literature, reports, and other publications not necessarily indexed in specialized databases. Notably, DuckDuckGo’s search algorithms vary from those of Google, which may present a more diverse set of results. To ensure a comprehensive search and to address potential biases, we meticulously screened the initial five pages of results from both Google and DuckDuckGo for pertinent content.

The search strategy was structured around two primary concepts: (1) COVID-19 and (2) health workers. Specific search terms included “health professional,” “health personnel,” and “health care worker”. The comprehensive list of keywords used for PubMed is detailed in Appendix A. For the purpose of this review, the term ‘health care worker’ (D006282) was defined broadly to include general and specialist medical practitioners, nurses, midwives, paramedical practitioners, dentists, pharmacists, physiotherapists, laboratory technicians, among others [18].

For the inclusion of the grey literature, we adhered to the same inclusion and exclusion criteria outlined in Table 1. Each piece of the grey literature was further evaluated by the authors to assess the reliability of their sources, ensuring that only credible and relevant materials were incorporated into our review.

Our search strategy for the grey literature was consistent with our approach to peer-reviewed articles. We utilized keywords on PubMed (as detailed in Appendix A), MedRxiv, Google, and DuckDuckGo that addressed the following two primary concepts: (1) COVID-19 and (2) health workers. To capture a broad range of the relevant literature, we employed search strings such as “health professional,” “health personnel,” and “health care worker.

### 2.3. Study Selection 

Search results were imported into the Covidence platform [19], duplicates were removed, and six independent reviewers screened the literature. Disagreements were resolved through discussion. Titles, abstracts, and full texts were screened based on specific inclusion and exclusion criteria (Table 1). The included studies were published in English, the primary research was available in its full text and were published after 1 December 2019, focusing on COVID-19 lessons regarding health worker readiness, resilience, psychological stress, or advances in training or clinical practice. Studies were excluded if innovations were planned but not implemented or if limited to clinical evaluations of COVID-19 therapies, diagnostics, or new vaccines. This scoping review focused on clinical education within the health workforce, while other reviews were also scheduled to be conducted using the other included studies. 

### 2.4. Data Charting

A data charting form was developed and used on the Covidence platform [19]. Data were charted according to aspects such as author, title, year of publication, country of study, study aims, study population (medical specialties), sample size, study design, key findings, recommendations (including actions), and recommendations for future studies.

### 2.5. Reporting the Results

A narrative synthesis of these findings was carried out, providing a detailed summary of the main themes and emerging trends observed within the included studies.

In synthesizing the data from the included studies, we recognized the inherent heterogeneity in terms of the study design, participant characteristics, and outcomes. Given this diversity, we opted for a narrative synthesis approach, which enabled us to descriptively summarize these findings without making direct statistical comparisons. This approach was deemed more suitable than a meta-analysis, which might have been inappropriate and potentially misleading due to the significant heterogeneity among these studies. It is essential to highlight that while narrative synthesis offers a comprehensive overview, it does not provide pooled effect estimates the same as meta-analysis. We believe that our approach, while qualitative, offers a holistic understanding of this topic. However, readers should be cautious when interpreting these findings, considering the inherent differences and potential biases in the primary studies. Future research with more standardized methodologies might allow for more quantitative synthesis methods.

## 3. Results

### 3.1. Overview

The primary database search yielded 10,323 articles, with 10,236 remaining after their duplicate removal. A two-step screening process (title and abstract, followed by the full text) resulted in the inclusion of 839 articles. Of these, 88 studies specifically addressed education and training for health workers during the COVID-19 pandemic and were incorporated into this scoping review. The included studies utilized quantitative, qualitative, and mixed-methods analyses (Figure 1).

### 3.2. Study Characteristics

This scoping review encompassed a total of 31,268 participants, including physicians, medical trainees, nurses, paramedics, students, and health educators. Notably, 11,656 participants were from a study conducted in Pakistan by Afzal et al. [20]. Most studies were carried out in high-income countries (n = 60; 68%), with fewer in upper-middle (n = 8; 9%) and lower-middle (n = 11; 13%) income countries. Nine studies (n = 9; 10%) were conducted across multiple countries with varying income levels, and three of these included low-income countries [21,22,23] (Table 2 and Figure 2).

A significant portion of these studies (n = 71, 81%) utilized cross-sectional designs. The majority (n = 77, 87.5%) employed quantitative analysis, while a smaller proportion (n = 11, 12.5%) used either mixed-method or qualitative analysis.

### 3.3. Impact of COVID-19 on Clinical Education

The multifaceted effects of COVID-19 on health workers’ training and education prompted investigations into the pandemic’s impact on clinical education and the integration of simulation training and virtual learning in the health sector. A total of 32 studies (36%) [20,23,24,25,26,27,28,29,30,31,32,33,34,35,36,37,38,39,40,41,42,43,44,45,46,47,48,49,50,51,52,53] assessed the consequences of COVID-19 on health workers’ clinical skills and training period. A small number of studies (n = 2, 6%) reported decreased motivation among medical and nursing students since the pandemic’s onset [24,25]. One study revealed that the negative impact on daily clinical education had led some medical trainees to contemplate changing careers due to a decline in confidence regarding their clinical skills [23].

#### 3.3.1. Impact on Clinical Skills and Abilities

Surgical and non-surgical training across various specialties and training levels were affected during the pandemic. Medical trainees experienced reduced exposure to percutaneous intervention procedures, endoscopies, radiological imaging readings, electrophysiology training, neurology duties, and the provision of pain medicine (n = 6, 19%) [26,27,28,29,30,31]. Trainees in general surgery, orthopedic surgery, ophthalmology, neurosurgery, otolaryngology, and plastic surgery experienced decreased surgical training and operative skills (n = 10, 31%) [32,33,34,35,36,37,38,39,40,41]. Pediatric urologists observed an 80–100% disruption in educational training [42], and gastroenterology trainees reported decreased involvement in specific procedures (30% in colonoscopy and 20% in esophagogastroduodenoscopy) [43]. Moreover, ophthalmology residents experienced a decline in surgical case numbers [44], and senior surgery residents in Pakistan completed fewer cases, with the number of minor and major cases dropping from 41 and 146 pre-COVID-19 to 11 and 40 cases, respectively, after the pandemic’s onset [45]. Radiology and pathology trainees were more concerned about missed educational opportunities compared to other specialties [46]. Gaps in educational training resulted from decreased trainee involvement in certain procedures and mentor unavailability [43]. Conversely, urology oncologists reported no change in trainees’ education during the pandemic; only senior urologists’ medical education was significantly affected due to reduced attendance at professional meetings [47].

#### 3.3.2. Impact on Clinical Training Period

First- and second-year residents felt disadvantaged concerning clinical training (n = 2, 6%) [38,48]. The majority of trainees (n = 12, 38%) were assigned to regular or COVID-19 wards [20,26,27,28,30,31,40,43,44,49,50,51], while most senior residents (n = 3, 9%) ceased attending clinics [33,34,36]. In total, 32–80% of medical trainees expressed concern about not meeting academic accreditation requirements and the potential need for extending their clinical training period (n = 9, 28%) [28,31,33,34,39,40,43,48,49]. In six academic medical centers in Boston, 75% of general surgery chief residents agreed to start their fellowship as scheduled without a delay in graduation, 16% proposed additional general surgery training during the fellowship program, and only 8% preferred delaying their graduation [52]. Community health workers requested adjustments to the training curriculum to cover more COVID-19-related health topics, such as prevention, clinical courses, community resources and engagement, vulnerable populations, mental health, and general COVID-19 information [53].

### 3.4. Adaptation Strategies

A total of 56 studies (64%) [20,21,54,55,56,57,58,59,60,61,62,63,64,65,66,67,68,69,70,71,72,73,74,75,76,77,78,79,80,81,82,83,84,85,86,87,88,89,90,91,92,93,94,95,96,97,98,99,100,101,102,103,104,105,106,107] examined various adaptation strategies to mitigate the impact of COVID-19 on clinical education and practice among health workers. Predominant strategies included simulation training, e-learning, and virtual training.

#### 3.4.1. Simulation Training

Fourteen studies (16%) [54,55,56,57,58,59,60,61,62,63,64,65,66,67] assessed the impact of introducing simulation training into the healthcare sector. These studies primarily targeted clinical procedures and interventions performed on COVID-19 patients in emergency departments or COVID-19 wards. Simulation training was found to increase interprofessional training relationships among health workers, improve clinical skills and technical performance, and serve as a mitigation and coping tool.

##### Interprofessional Training Relationships among Health Workers

Bode et al. [54] reported an increased awareness of the roles of different occupational groups within healthcare teams, including nursing trainees and medical students, as a result of conducting an interprofessional simulation course in Germany. Similarly, multi-professional simulation training in Pakistan involving nurses, doctors, sanitation workers, laundry service workers, and ambulance drivers improved participants’ preparedness in managing patients with COVID-19 infection under strict isolation measures [55]. The simulation training incorporated both theoretical and hands-on sessions, during which participants practiced clinical procedures using mock patients and manikins. Covered topics included the use of personal protective equipment (PPE), PCR testing, obtaining intravenous blood samples, disinfecting and cleaning infectious fluids via sanitation staff, transporting COVID-19 patients, and disinfecting ambulances [55].

##### Clinical Skills and Technical Performance

Various forms of simulation training were implemented, resulting in predominantly positive outcomes, with health workers feeling more confident and prepared after training. Simulation training for nasopharyngeal swabbing for COVID-19 significantly increased procedural competency among 46 health workers, with an average increase of 1.41 points (from 3.13 to 4.54) [56]. Participants practiced nasopharyngeal swabbing using a high-fidelity airway simulation model following a brief lecture.

Simulation training for COVID-19 airway management improvement led to increased comfort and enhanced clinical skills during intubation [57,58]. Trainees, nurses, physicians, and respiratory therapists engaged in simulations using an adult or a six-year-old-advanced patient simulator while wearing gowns, goggles, and shields due to PPE shortages [57]. Munzer et al. developed an airway algorithm, which was applied by healthcare professionals using Styrofoam masks and replicating N95 masks (due to PPE shortage) in an in situ simulation involving a decompensating COVID-19 patient requiring intubation [58].

The worldwide implementation of structured simulation training and debriefing for pediatric anesthesia staff across 39 institutions resulted in participants feeling better prepared to manage COVID-19 patients [59]. Dharamsi et al. evaluated the use of in situ simulation programs followed by debriefing sessions in Canadian emergency departments. A modified manikin that aerosolized phosphorescent droplet secretions was used, with secretions visualized on providers at the end of the simulation using black light. Most participants (97%) confirmed the simulation training’s relevance to their practice, and 94% felt better prepared and ready to care for COVID-19 cases in emergency departments [60].

In Denmark, non-intensivist doctors underwent theoretical sessions followed by hands-on training in mechanical ventilation, hemodynamic monitoring, vascular access, and PPE donning and doffing. This training led to the improved acquisition and retention of newly learned techniques [61]. However, German dentistry students performed worse on a simulated state examination compared to actual patients, with no significant difference in structured theoretical examinations [62].

PPE usage improved after in situ simulation training [63,64]. Pediatric healthcare providers worldwide implemented COVID-19 simulation training, primarily conducting PPE training through videos and in situ training, while airway management and cardiopulmonary resuscitation were also delivered [64]. Telesimulation was also initiated for COVID-19 education, replacing live-simulation training in some cases. Another study revealed that simulation-based educational intervention on PPE donning and doffing raised performance scores from 2.5 points pre-test to 7.9 points post-test. In terms of performance and cognitive load, most participants moved from the inefficient quadrant (low performance—high cognitive load) pre-test to effective (high performance—high cognitive load) and the efficient (high performance—low cognitive load) quadrants post-test [65].

##### Mitigation and Coping Tool

Simulation training has proven to be effective in preparing staff to manage the cognitive load associated with caring for COVID-19 patients [59]. In a study conducted in Colombia, 54.5% of participants (n = 33) agreed that a simulated intervention in PPE donning and doffing reduced individual and collective stress [65]. In the United States, a structured train-the-trainer simulation program targeting PPE usage and airway management skills among emergency and critical care physicians significantly increased comfort levels from 2.93 to 4.35 when managing COVID-19 patients [66]. A Saudi Arabian study found that 57.4% of emergency department healthcare professionals felt more comfortable dealing with unstable COVID-19 patients after implementing mock codes (simulation drills) involving PPE use and basic and advanced airway techniques [67].

#### 3.4.2. E-Learning and Virtual Training

In 2020, resources were allocated to online and web-based educational activities due to the COVID-19 pandemic [68]. A total of 42 studies (48%) [20,21,69,70,71,72,73,74,75,76,77,78,79,80,81,82,83,84,85,86,87,88,89,90,91,92,93,94,95,96,97,98,99,100,101,102,103,104,105,106,107] examined the introduction of virtual learning into health workers’ education and its impact on clinical practice. The e-learning modalities comprised online courses, educational mobile applications, multimedia training videos, social media educational posts, webinars, and virtual conferences.

##### E-Learning in Daily Clinical and Surgical Practice

The positive impact of a web-based module on health workers’ adherence to hygiene measures in a Pakistani hospital, utilizing CDC and WHO guidelines for a 20–30 min asynchronous course on hand and respiratory hygiene techniques, was demonstrated by Abbas et al. [69]. In an Australian private clinic, physiotherapists recognized e-learning as comprehensive, self-paced, and beneficial for patient care management, employing both synchronous and asynchronous learning, including mock video consultations and pilot patients [70]. A Swiss academic hospital analyzed an mHealth platform’s usage, which provided easy access to validated medical content, observed a significant increase in active devices and daily user activity during the COVID-19 peak [71].

Plastic and general surgery trainees in Italy used multimedia training videos offered by multiple online sources to prepare for oncologic, oncoplastic, and reconstructive breast surgeries, as reported by Marcasciano et al. [72]. Gastrointestinal and endoscopic surgeons observed increased memberships and activity in closed Facebook groups during the pandemic [73]. Neurosurgeons, orthopedists, and radiologists value teleconferencing for spine education, with presenters delivering synchronous presentations to international participants via Zoom [74].

Virtual educational training programs improved the operational assessment and practices of healthcare providers in surgical emergency departments and assisted living facilities [75,76]. Nurses in a Chinese emergency department attended WeChat-based training sessions, covering COVID-19-related knowledge and practices [75]. Canadian long-term care home healthcare providers participated in a modified ECHO project involving weekly virtual sessions and resource sharing [76]. Virtual learning positively impacted first-year health students’ abilities to conduct in-person interviews with older individuals, with 94.8% of surveyed students appreciating the introduction [77].

Theoretical education compensated, to some extent, for the clinical training shortages caused by COVID-19 [78]. Trainees engaged in research projects and online educational activities, though a decrease in the application of learned knowledge in practice was observed [79]. Four studies noted increased self-reported and objective knowledge of disaster preparedness following virtual training [80,81]. Participants agreed on the continued importance of webinars for clinical practice post-pandemic [82,83].

##### Students’ Support in Generating and Managing Web-Based Lectures

E-scouts, digitally adept medical and dentistry students at a German university, assisted lecturers in managing online platforms during web-based classes. They aided in implementing case-based e-learning, preparing audio commentaries and video recordings of presentations, and receiving positive feedback for their digital skills and prompt responses [84]. Another German study emphasized the importance of student–teacher collaboration in developing online lectures and seminars, with the successful digitalization of the ‘interactive training for clinical decisions’ module using the plan-do-check-act (PDCA) cycle and well-received video lecture substitutes [85].

##### Increased Global Reach and Communication via Virtual Platforms

Virtual training improved interactivity, outreach, and flexibility [82]. Swords et al. described an international, multidisciplinary virtual tracheostomy education in which participants established a successful online tracheostomy-care network consisting of a web-based platform and five one-hour webinar sessions, receiving positive feedback on course length, applicability, and expertise [80]. In separate studies, fellowship applicants for advanced gastrointestinal minimally invasive surgery (MIS) and Complex General Surgical Oncology (CGSO) programs reported positive experiences with virtual interviews using Zoom, although in-person interviews allowed for a better evaluation and ranking of the CGSO program [86,87].

In 2020, pharmacy students participated in a 9-week online Advanced Pharmacy Practice Experience (APPE) instruction program instead of a traditional clinical rotation, with 81% reporting a positive impact on their virtual learning experience due to collaboration with diverse preceptors [88]. The Virtual Grand Rounds (VGR) project involved urology applicants presenting during scheduled grand rounds via Zoom or WebEx. Applicants found this method effective for learning about outside programs, while 50% of faculty participants felt confident in assessing candidates [89]. In the United States, an online platform was created to inform oncologists regarding practice changes during the COVID-19 pandemic, with 47% of oncologists nationwide viewing the website over two months [90].

##### Perceptions of Virtual Education

During the COVID-19 pandemic, health workers, trainees, students, and educators reported a mix of positive and negative experiences and perceptions regarding virtual education.

##### Perceptions among Health Students and Educators

Virtual education was delivered online through webinars, conferences, and continuous medical education activities [91]. In the UK, medical students found online Small Group Teaching (SGT) to be as effective as face-to-face SGT, with 335 students expressing satisfaction [92]. Similarly, nursing students experienced an increase in their Health Education Systems, Inc (HESI) scores in Fundamentals, while HESI scores in Maternity, Psychiatric, and Medical Surgical Nursing remained unaffected [93]. A 90 min workshop in the United States employed the Zoom platform to provide instructional materials and introduce health educators to relevant learning theories and interactive teaching tools both within and outside of Zoom. Participants were given opportunities to practice using these tools, and after engaging in small-group discussions, health educators reported feeling comfortable using various interactive tools for online education [94]. Medical faculty members in Saudi Arabia observed that evaluations of virtually delivered courses with structured feedback resulted in high self-perceived competency scores among health educators [95]. Medical training lecturers at a German university noted that the COVID-19 pandemic served as a catalyst for transitioning to virtual medical education, facilitating the implementation of new teaching methods such as online lectures, collaborative working, live broadcasts, and online chats [96]. Researchers at a German medical school found that in-person and virtual career counseling sessions were equally well-received [97]. Overall, program directors and deans of medical schools expressed approval and satisfaction with the quality of existing online medical education [98,99,100].

By contrast, nursing students and faculty members in Canada held a negative view of online education due to a lack of practical experience [101]. Although nursing faculty members at an American university demonstrated strong self-efficacy during online teaching, they observed poor student engagement [102].

##### Perceptions among Health Workers and Trainees

In Indonesia, health professionals, including physicians, nurses, and pharmacists, expressed high satisfaction with a webinar series as a form of continuing education [103]. The series comprised six webinars delivered over two consecutive days using the Zoom platform and YouTube Live, addressing COVID-19 and non-COVID-19 medical topics such as medicolegal aspects of medicine, metabolic disorders, neurological disorders, and emergency cases from various organ systems. Medical trainees favored online discussions over traditional face-to-face instruction and advocated for the incorporation of virtual faculty exchange sessions throughout the academic year, citing their flexibility and usefulness [104,105]. The virtual educational program also mitigated environmental impacts, reducing airfare costs (>15,000 USD) and carbon emissions (>24 metric tons) [105]. Neurosurgeons from different countries also called for a digital transformation of conferences and scientific meetings to decrease travel expenses [20]. German general practitioners participated in an e-learning program based on a vocational training course for general medical practice, which was established in 2017 and initially conducted through in-person seminars. The e-learning program consisted of six synchronous and two asynchronous 45 min units, with participants expressing satisfaction regarding time-saving, cost-effectiveness, and the adaptability of the online experience [106].

Conversely, Ismail et al. reported that physicians across various specialties felt overwhelmed by the sheer number of webinars during the COVID-19 pandemic, suggesting the need for guidelines and regulations for web-based meetings [21]. An excessive number of webinars, repetitive information, and low participant engagement led to stress among ophthalmologists in a study conducted in India [83]. Additionally, web-based education faced challenges in providing appropriate personal interactions with peer groups due to technological issues, unstable internet connections, and insufficient computer facilities [106,107] (Table 3 and Table 4).

## 4. Discussion

This scoping review analyzed 88 studies, providing valuable insights into the impact of COVID-19 on medical education and training, as well as the effectiveness of adopting alternative approaches, such as simulation training and virtual learning. This review reveals that the pandemic led to reduced exposure to surgical and interventional skills, diminished patient interactions, and increased concern among trainees about completing their clinical education on time. 

Gaps in educational training were caused by decreased involvement in procedures, the unavailability of mentors, and deployment to cover COVID-19 wards. Simulation training, e-learning, and virtual training were the most common approaches used to minimize the impact of COVID-19 on clinical education. 

The future competence of medical trainees in their chosen medical fields largely depends on the quality of their clinical training [108]. During pandemics, medical trainees are often deprioritized in terms of patient care, resulting in limited exposure to clinical settings [109,110]. The COVID-19 pandemic significantly limited trainees’ exposure to surgical procedures, hands-on intervention skills, and face-to-face patient interactions [26,27,28,29,30,31,32,33,34,35,36,37,38,39,40,41,42,43,44,45,46,47]. In-hospital educational floor rounds for junior trainees and outpatient clinics for senior trainees [33,34,36,38,48] were severely impacted and almost ceased entirely. Consequently, early-stage residents were frequently deployed to assist overburdened healthcare facilities with their COVID-19 responses [20,26,27,28,30,31,40,44,49,50,51]. These disruptions heightened students’ and trainees’ concerns about completing their clinical education on time and obtaining academic qualifications [33,34,40,43,48,52,53].

The use of in situ simulation in healthcare education began in the early 2000s and targeted the training of health workers in managing critical patient situations in emergency departments [111]. Over time, the use of simulation training reached other clinical settings, including in-hospital and primary care clinics [111,112]. Sørensen et al. discussed the different simulation settings such as off-site (in simulation centers or in training rooms within the hospital) and in situ simulations (in clinical settings). All simulation settings led to enhanced individual and team learning [113]. During the pandemic, in situ simulation training emerged as a crucial tool for healthcare professionals, providing a flexible and adaptable instructional method. This approach facilitated the swift adoption of new protocols and practices, allowing redeployed health workers to rapidly acclimatize to novel roles. Interprofessional simulation training was particularly beneficial, fostering a deeper understanding of each professional’s responsibilities and ultimately improving the quality of patient care. Simulation training catered to a diverse population of health workers involved in COVID-19 case management, including intensivist and non-intensivist physicians, nurses, respiratory therapists, anesthesiologists, and emergency department staff [59,60,61,65]. Interprofessional simulation training allowed health workers to gain better insights into each other’s roles and management tasks [54,55], suggesting that hospitals should consider incorporating this approach into regular training programs to enhance patient care. These reviewed studies revealed that simulation training boosted health workers’ confidence, particularly when learning new skills, such as COVID-19 testing, airway management techniques, stabilizing COVID-19 patients, and donning and doffing PPE, which could also be delivered through video-based simulation training [59,60,61,65]. Other than providing a platform for teamwork and individual clinical growth, as was discussed in this review, simulation training offers a safe learning environment for health workers to practice clinical skills without harming real patients [113,114]. It also allows the same clinical scenario to be practiced multiple times while obtaining immediate feedback from instructors [114,115]. However, the high cost of the specialized equipment and the limited reproducibility of the simulation training to real-life situations due to personal and environmental factors are considered drawbacks [114]. Hence, further research is warranted to assess the long-term impact of simulation training on healthcare professionals’ readiness for real-life clinical situations.

Virtual learning served as an essential component in sustaining clinical education during the pandemic, enabling the dissemination of knowledge among healthcare professionals, supporting research activities, and fostering global engagement through online conferences and webinars. While virtual learning offers several advantages, it also presents challenges that must be addressed to optimize the experience, including limited opportunities for hands-on practice, an abundance of virtual events, and unreliable internet connections. Virtual learning has become essential for not only the acquisition of theoretical knowledge but also for developing clinical and operational skills. It has resulted in improved preparedness for surgeries, adherence to hygiene guidelines, and enhanced patient care management skills [69,70,71,72,73,74,75]. The accessibility of clinical education, facilitated by mobile applications and online platforms, has enabled knowledge dissemination among healthcare workers without the constraints of a physical environment or specific timings. Virtual learning is regarded as an effective means to enhance knowledge and provide guidance for medical trainees, whether through online medical lectures or web-based career counseling sessions [96,97]. As per the literature, virtual learning expressed the same benefits and drawbacks in the pre-COVID-19 era as during the pandemic. Virtual learning exhibited convenience, cost-effectiveness, and accessibility and facilitated the transmission of knowledge. However, poor internet connectivity, limited access to technology, and a lack of interactivity were also identified [116]. Further research is needed to address innovative measures that can serve as solutions for the different challenges presented. 

Teleconferences and webinars were established as a useful and beneficial method of continuous education in both medical and surgical specialties before the onset of the COVID-19 pandemic [108,117,118,119,120,121]. When access to specialty training, such as in-person training for surgical and percutaneous intervention skills, became severely limited, it was crucial to involve students in specialty-specific simulation training [122] alongside the acquisition of theoretical knowledge. Virtual learning was not recognized, however, as a significant method for clinical education until the onset of the pandemic [108]. During the COVID-19 pandemic, trainees engaged in research activities, webinars, conferences, and online meetings on a regular basis to help offset the reduction in face-to-face learning experiences. They participated in research activities related to COVID-19 or their chosen specialty [23,30,31,41,48]. Involvement in research projects allowed trainees to maintain their academic education while contributing to the medical knowledge base. 

The pandemic-induced shift to virtual learning environments underscored the potential for the further incorporation of digital tools into medical education curricula. Effective online medical education necessitates student–lecturer interactions and structured feedback from health educators [84,85,95]. Studies showed that strengthening the student–lecturer interaction requires an enhanced sense of community during online lectures; engaging in group discussions can compensate for the lack of physical presence [123,124]. Working in small group sizes, performing online quizzes, and participating in online case simulations also enhance student interaction and engagement [125].

Online conferences and webinars not only enhanced participants’ knowledge and skills but also facilitated greater global reach and interaction while adhering to COVID-19 social distancing guidelines [80,81,82,83,86,87,88,89,90]. Webinars simplified the sharing of up-to-date guidelines on COVID-19 management and the dissemination of clinical changes across various specialties.

The majority of health educators, professionals, and students reported high satisfaction levels with virtual learning [92,93,98,99,100,103,104,105]. Additional benefits included time-saving, flexibility, reduced travel expenses, and decreased carbon emissions [21,105,106], laying the groundwork for the expansion of virtual learning across academic and clinical settings. Although virtual learning effectively refined theoretical and clinical knowledge while maintaining safety measures during the pandemic, several drawbacks emerged. Addressing issues such as limited hands-on practice, restricted participant involvement, inadequate internet connections, and an excessive number of virtual events and online meetings is essential to enhance the virtual learning experience [22,83,101,102,106,107].

In conclusion, this scoping review suggests that virtual learning, simulation training, and research activities have played a crucial role in mitigating the impact of COVID-19 on medical education and training. However, there are still gaps in our understanding of the long-term effects of these approaches on trainees’ clinical competency and overall preparedness. Future research should focus on addressing these identified drawbacks, exploring innovative methods to enhance the medical education experience, and evaluating the long-term outcomes of these alternative strategies on healthcare professionals’ clinical performance.

### Limitations

In conducting this review, there may have been an inadvertent emphasis on particular healthcare institutions, clinical specialties, or geographic regions, potentially limiting the generalizability of our findings. Given the rapidly changing landscape of the COVID-19 pandemic, this review might not have fully captured the entirety of its impact on clinical education. Additionally, while PubMed was our primary academic database due to the depth and breadth it offers on this topic, we acknowledge that relying predominantly on one database might have led to the inadvertent exclusion of some relevant studies.

## 5. Conclusions

The COVID-19 pandemic significantly impacted the education and training of health workers across various sectors and training levels. Health workers were redirected to COVID-19 wards, which probably had an impact on their skills and knowledge in specific fields. Consequently, their training periods may need to be extended to ensure high-quality care for patients upon their return to their original roles. Simulation training emerged as a beneficial approach, enhancing interprofessional learning, improving hands-on clinical skills in relation to COVID-19 management, and serving as a coping and mitigation tool. Effective simulation training often begins with an introductory lecture or video to review essential concepts and techniques, incorporates high-fidelity simulation models or mock patients, and concludes with debriefing sessions to discuss participant performance.

E-learning, encompassing online courses, mobile apps, webinars, and virtual conferences, proved valuable and constructive for daily clinical and surgical practice, as well as non-clinical areas. Virtual learning facilitated greater global reach and knowledge exchange on an international scale. Several effective e-learning approaches, including synchronous and asynchronous online programs, mobile health platforms, surgical training videos, teleconferencing, and continuous education through virtual learning, were employed to equip health workers with COVID-19-related medical knowledge.

Proposals for enhancing clinical education and training include incorporating crisis education and simulation teaching into medical curricula, evaluating the impact of virtual learning on future clinical practice, assessing the effects of mobile health platforms on guideline adherence and patient outcomes, addressing COVID-19-related clinical gaps on trainee readiness, refining online educational tools, implementing train-the-trainer strategies for high-risk interventions, and developing internet-based curricula for continuing medical education. Furthermore, there is a need to evaluate the effectiveness of virtual faculty exchange programs and remote clinical conferences, assess emergency department mock code impacts on patient outcomes, and improve professional interactions during e-learning in vocational training.

Several challenges in implementing online education and training programs have been identified, including a lack of practical experience, poor student engagement, excessive webinars with repetitive information, unstable internet connections, and insufficient personal interactions among peer groups. Further research is necessary to enhance preparedness for future pandemics or similar situations and to address these challenges in the development and implementation of online education and training programs.

## Figures and Tables

**Figure 1 healthcare-11-02902-f001:**
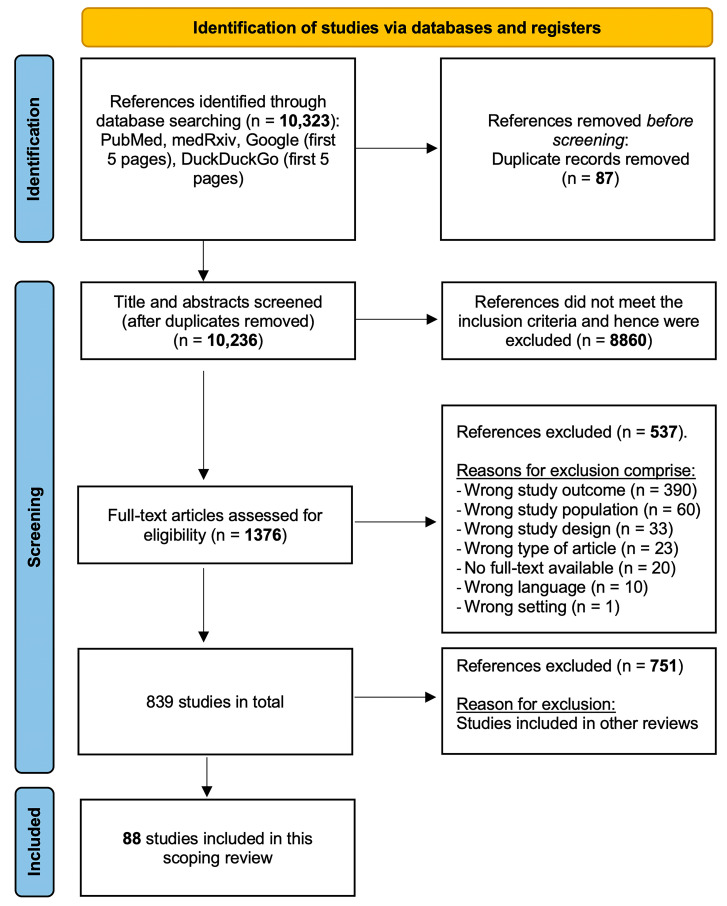
PRISMA-ScR flowchart.

**Figure 2 healthcare-11-02902-f002:**
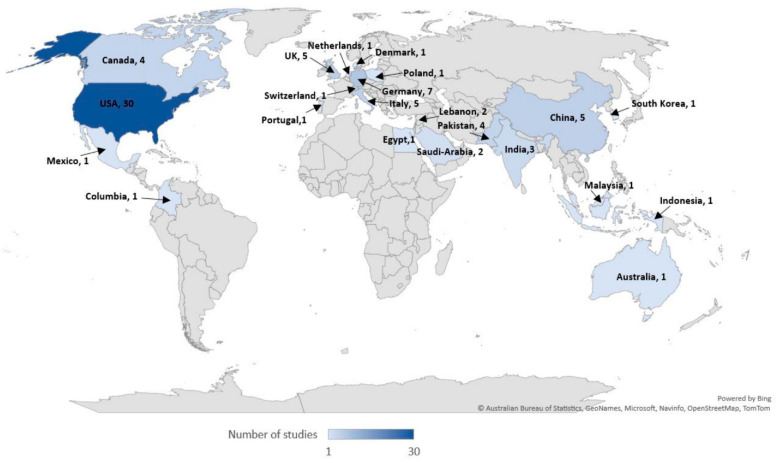
Included studies per country.

**Table 1 healthcare-11-02902-t001:** Inclusion and exclusion criteria.

Inclusion Criteria	Exclusion Criteria
- General: Published in English Full-text available Published after 1 December 2019 Any primary research Grey literature - Population: health workers (physicians, nurses, healthcare students, pharmacists, dentists, physical therapists, speech therapists, radiologic technologists, medical laboratory technicians, mental health professionals, community health workers, public health workers, nutritionists and dietitians, paramedics) - Focus: COVID-19 impact, COVID-19 preparedness and learnings, strategies for better preparedness for major public health emergencies, interventions at the community level, early detection methods, surge response, resilience among health workers, workload and psychological burden, mental health and burnout, innovations in training and care models, innovations in mobilization, interprofessional collaboration	- Study type: commentary, editorial, review, modeling study, study protocol, case study - Focus: biological/microbiological evaluation, COVID-19 diagnostics, vaccine efficacy, innovations without certain validation, incidence/prevalence rate, patient focus, antibody screening/prevalence, mask efficacy

**Table 2 healthcare-11-02902-t002:** Overview of study characteristics.

Study Characteristics	Studies, n (%)		Participants, n (%)	
	88 (100)		31,268 (100)	
**Countries**				
	**Studies (n)**	**% of All Studies**	**Participants (n)**	**% of All Participants**
High-income	60	68	10,010	32
Australia	1	1.1	15	0.05
Canada	4	4.5	512	1.6
Denmark	1	1.1	54	0.2
Germany	7	8	476	1.5
Italy	5	5.7	790	2.5
Korea	1	1.1	37	0.1
Netherlands	1	1.1	317	1
Poland	1	1.1	80	0.3
Portugal	1	1.1	75	0.2
Saudi Arabia	2	2.3	792	2.5
Switzerland	1	1.1	1233	3.9
UK	5	5.7	997	3.2
US	30	34	4632	14.8
Upper middle income	8	9	1006	3.2
China	5	5.7	552	1.8
Columbia	1	1.1	61	0.2
Malaysia	1	1.1	100	0.3
Mexico	1	1.1	293	0.9
Lower middle income	11	13	17,603	56.3
Egypt	1	1.1	346	1.1
India	3	3.4	1111	3.6
Indonesia	1	1.1	3607	11.6
Lebanon	2	2.3	321	1
Pakistan	4	4.5	12,218	39
Multinational	9 (of which 3 are in low-income countries)	10	2649	8.5
Study design				
Cross-sectional	71	81	28,658	91.7
Longitudinal interventional	15	17	2490	8
Randomized controlled trial	1	1	120	0.3
Longitudinal observational	1	1	-	-

**Table 3 healthcare-11-02902-t003:** Strengths and weaknesses of simulation training and e-learning implemented to address the effects of COVID-19 on clinical education among health workers.

	Simulation Training	E-Learning
Strengths		
Knowledge	- Increased comprehension and awareness of the occupational roles of other health workers [54,55] - Enhanced knowledge retention about stabilizing patients with COVID-19 [61]	- Augmented specialty-directed and COVID-19 medical knowledge among health workers [70,80,81,88,93] - Facilitated accessibility and dissemination of medical knowledge and information via mobile health platforms, closed Facebook groups, YouTube, scientific portals, Zoom, and WebEx platforms [71,72,73,74,90,91,92,96]
Skills	- Enhanced clinical skills related to COVID-19 management, including PPE usage, PCR testing, blood sampling, and airway management [55,56,57,58,61,63,64,65]	- Improved clinical and surgical patient management abilities, incorporating COVID-19 safety measures [70,75,76] - Increased adherence to specific hand and respiratory hygiene techniques [69]
Resilience	- Greater comfort and reduced stress when using PPE and managing airways in COVID-19 patients [57,58,59,60,65,66,67]	- Adaptation and adjustment to novel learning platforms and techniques [23,30,31,32,41,48,77,78,80,82,83,84,85,86,87,89,94,95,97,98,99,100,103,104,105]
Confidence	- Enhanced ability to manage COVID-19 patients with increased assurance [56]	
Practicality		- Time saving, reduction in travel expenses, and decreased carbon emissions [21,105,106]
Weaknesses		
Skills	- Diminished performance in simulated dentistry exams compared to real-life scenarios involving actual patients [62]	- Reduced application of theoretical knowledge to practical clinical settings [79,101]
Engagement and interaction		- Suboptimal participant engagement and interaction with peer groups [83,102,106,107] - Overabundance of webinars and repetition of information [22,83] - Technological issues and lack of stable internet connection [106,107]

**Table 4 healthcare-11-02902-t004:** Recommendations for enhancing education and training of health workers.

Overarching Themes	Recommendations
Innovation and Adaptation	- Develop innovative programs to enhance trainees’ education and compensate for the reduced number of procedures performed during the pandemic [31,36,43,49] - Incorporate crisis education and simulation teaching into medical educational systems to prepare for future pandemics and public health threats [58,61,62,64] - Embrace web-based conferences and webinars for their cost-effectiveness, accessibility, time-saving, and environmental benefits [21,74,88,90,106]
Virtual learning	- Engage trainees in virtual clinical work and provide them with research opportunities [32,33,37,46,48] - Implement virtual learning in fellowship interviews, mentoring programs, the acquisition of clinical procedures, disaster preparedness, and medical education curricula as a complement to traditional in-person teaching [72,73,76,77,80,81,82,86,87,89,92,93,97,99,103] - Evaluate the impact of virtual learning on the quality of clinical practice and incorporate virtual case-based teaching into the curricula of health professions as a contingency plan or to facilitate the adoption of telehealth in future clinical practice [68,69,70]
Simulation Training	- Integrate simulation training into hospitals and medical centers to improve health workers’ readiness, effectiveness, and confidence when dealing with new and high-risk procedures, including donning and doffing PPE [55,56,59,63,65,67] - Enhance team communication through multi-professional simulation training to optimize the quality of patient care [54,57,60].
Program Evaluation and Improvement	- Assess trainees’ preparedness for independent practice and extend the training program when necessary [28,39,40,45,52] - Establish proper guidelines and regulations for web-based meetings to improve personal interactions, address technical difficulties, reduce inconvenient timing, minimize overlapping topics and biased scientific content, and help health educators and students become more familiar with online platforms [22,83,85,94,95,96,98,100,101,102]

## Data Availability

No new data were created or analyzed in this study. Data sharing is not applicable to this article.

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
