# Peer review of "Education and Training Adaptations for Health Workers during the COVID-19 Pandemic: A Scoping Review of Lessons Learned and Innovations"

_healthcare, 2023, doi:10.3390/healthcare11212902_

Round 1
Reviewer 1 Report
Comments and Suggestions for Authors
This review highlights the significant impact of the COVID-19 pandemic on the clinical education and training of health workers worldwide. It highlights the disruptions that healthcare facilities faced and the limitations in acquiring new clinical skills during the pandemic. To address these challenges, the paper discusses the implementation of adaptation measures, including simulation training and e-learning. The results of the review indicate that, in some cases, the pandemic's effects on health workers' clinical skills necessitated the extension of training periods. Nevertheless, the adoption of simulation training and e-learning as adaptation strategies yielded several positive outcomes, including improved technical and clinical performance, increased confidence, and an expanded global educational outreach.
Regarding the topic, this manuscript addresses a significant area and appears to be informative. The questions addressed are clearly stated, and the research design is appropriate. Overall, the manuscript provides an organic overview, with a densely organized structure and based on well-synthetized evidence, it is clear and easily to read. The abstract is comprehensive and well-structured. The introduction provides detailed background with relevant and appropriate literature. Methods, results, and discussion sections are divided into paragraphs and complemented by detailed figures and tables, which enhance comprehension. The discussion properly debates the impact of COVID-19 on medical education and the importance of alternative approaches. The conclusion effectively summarizes the main points of the text and offers valuable recommendations for enhancing health worker education and training, especially when considering the challenges presented by the COVID-19 pandemic.
The following minor rephrasing suggestions aim to improve the clarity and flow of the text while maintaining its informative and well-structured nature.
Lines 145-146
In the sentence "A few studies (n=2, 8%) reported that medical and nursing students were less motivated to study since the onset of the pandemic [24,25]," consider rephrasing to improve clarity, e.g., "A small number of studies (n=2, 8%) reported decreased motivation among medical and nursing students since the pandemic's onset [24,25]."
Lines 146-148
In the sentence "In one study, the negative impact on daily clinical education led some medical trainees to consider changing careers due to diminished confidence in performing clinical skills [23]," consider rephrasing for clarity, e.g., "One study revealed that the negative impact on daily clinical education had led some medical trainees to contemplate changing careers due to a decline in confidence regarding their clinical skills [23]."
Lines 174-176
Maintain consistency in punctuation and phrasing for improved readability, e.g., "Most trainees were deployed to regular wards or COVID-19 wards (n=12, 35%) [20,26–28,30,31,40,43,44,49–51], and the majority of senior residents stopped attending clinics (n=3, 9%) [33,34,36]." can be revised as "The majority of trainees (n=12, 35%) were assigned to regular or COVID-19 wards [20,26–28,30,31,40,43,44,49–51], while most senior residents (n=3, 9%) ceased attending clinics [33,34,36]."
Lines 405-407
Consider revising "During pandemics, medical trainees are deprioritized from attending patient care which results in limited exposure to clinical settings [110,111]." for clarity, e.g., "During pandemics, medical trainees are often deprioritized in terms of patient care, resulting in limited exposure to clinical settings [110,111]."
LINES 515-516
Consider revising "Health workers were redirected to COVID-19 wards, which likely affected their skills and knowledge in their respective fields" for clarity, e.g., "Health workers were redirected to COVID-19 wards, which probably had an impact on their skills and knowledge in their specific fields."
Lines 537-538
In "Moreover, efforts should be made to evaluate virtual faculty exchange programs and remote clinical conferences," consider rephrasing for clarity, e.g., "Furthermore, there is a need to evaluate the effectiveness of virtual faculty exchange programs and remote clinical conferences."
I thoroughly enjoyed reviewing this manuscript, and I would like to congratulate the authors because they carried out a valuable investigation that may represent a considerable contribution to the related research.
Reviewer 2 Report
Comments and Suggestions for Authors
The provided study titled "Education and Training Adaptations for Health Workers during the COVID-19 Pandemic: A Scoping Review of Lessons Learned and Innovations" has several notable issues that need improvement:
- Introduction: The introduction lacks clarity and conciseness. It briefly mentions the impact of the COVID-19 pandemic on health worker education and training but does not effectively provide context or state the specific objectives of the study. A more comprehensive introduction is needed to set the stage for the research.
- Methods: The methods section is incomplete and lacks important details. It mentions using the PRISMA-ScR framework and Arksey and O'Malley's methodological guidance but doesn't describe how they were applied. Critical information such as inclusion and exclusion criteria for studies, quality appraisal methods, and the approach to data analysis is missing.
- Data Analysis Methods: The study does not specify the methods used for data analysis. It is unclear whether the review employed a narrative synthesis or another approach. This lack of clarity affects the reader's understanding of how the results were synthesized.
- Database Selection: The study mentions searching PubMed, medRxiv, Google, and DuckDuckGo databases but doesn't explain the rationale behind this selection. There should be a clear justification for the choice of databases, and the study should address the potential limitations of this selection in terms of comprehensiveness.
- Heterogeneity: The study appears to have merged participants and other characteristics from various studies but did not conduct a meta-analysis or address heterogeneity. This omission raises questions about how the data from different studies were synthesized and whether it was appropriate to do so without considering heterogeneity.
- Grey Literature: The study mentions including grey literature but does not provide details about the search strategy or the criteria used to include or exclude grey literature. This omission makes it challenging to evaluate the comprehensiveness of the review.
- Results: The results section appears to include studies without proper citation or reference, as indicated by the example on page 23, line 143 "26 studies assessed the consequences of COVID-19". This lack of clarity makes it difficult for readers to verify the validity of the information presented. To avoid long lines of references, these can be included in an appendix.
In summary, the study has several methodological and reporting issues that need to be addressed to enhance its rigor and transparency. Improving the clarity of the introduction, providing complete details in the methods section, clarifying the data analysis approach, justifying the database selection, addressing heterogeneity, and ensuring proper referencing are essential steps to improve the quality and credibility of the research.
Author Response
Kindly see the attachment.

Round 2
Reviewer 2 Report
Comments and Suggestions for Authors
The authors have demonstrated a commendable response to the provided feedback, addressing the comments appropriately and significantly enhancing the introduction, methods, and results sections of the manuscript.
The introduction now offers a more engaging foundation for the research, drawing the reader into the significance of the study and the key issues at hand. The methods section has been refined, and the skeleton outline is thoughtfully structured, guiding the reader through the research process. Similarly, the results section has been significantly improved.
However, it is important to note that a limitation section is notably absent in the revised manuscript. It remains essential to acknowledge the boundaries and potential shortcomings of the study, as this lends credibility and transparency to the work.
Author Response
Kindly see the attachment.
